# Performance Evaluation of Machine Learning Algorithms for Sarcopenia Diagnosis in Older Adults

**DOI:** 10.3390/healthcare11192699

**Published:** 2023-10-09

**Authors:** Su Ozgur, Yasemin Atik Altinok, Devrim Bozkurt, Zeliha Fulden Saraç, Selahattin Fehmi Akçiçek

**Affiliations:** 1Department of Biostatistics and Medical Informatics, Faculty of Medicine, Ege University, 35040 Izmir, Turkey; 2Translational Pulmonary Research Center—EgeSAM, Ege University, 35040 Izmir, Turkey; 3Department of Pediatric Endocrinology, Faculty of Medicine, Ege University, 35040 Izmir, Turkey; yaseminatik@yahoo.com.tr; 4Department of Internal Medicine, Faculty of Medicine, Ege University, 35040 Izmir, Turkey; devrim.bozkurt@ege.edu.tr; 5Division of Geriatrics, Department of Internal Medicine, Faculty of Medicine, Ege University, 35040 Izmir, Turkey; fulden.sarac@ege.edu.tr (Z.F.S.); fehmiakcicek@gmail.com (S.F.A.)

**Keywords:** predictive preventive personalized medicine (3P/PPPM), sarcopenia, machine learning, early diagnosis, older adults

## Abstract

Background: Sarcopenia is a progressive and generalized skeletal muscle disorder. Early diagnosis is necessary to reduce the adverse effects and consequences of sarcopenia, which can help prevent and manage it in a timely manner. The aim of this study was to identify the important risk factors for sarcopenia diagnosis and compare the performance of machine learning (ML) algorithms in the early detection of potential sarcopenia. Methods: A cross-sectional design was employed for this study, involving 160 participants aged 65 years and over who resided in a community. ML algorithms were applied by selecting 11 features—sex, age, BMI, presence of hypertension, presence of diabetes mellitus, SARC-F score, MNA score, calf circumference (CC), gait speed, handgrip strength (HS), and mid-upper arm circumference (MUAC)—from a pool of 107 clinical variables. The results of the three best-performing algorithms were presented. Results: The highest accuracy values were achieved by the ALL (male + female) model using LightGBM (0.931), random forest (RF; 0.927), and XGBoost (0.922) algorithms. In the female model, the support vector machine (SVM; 0.939), RF (0.923), and k-nearest neighbors (KNN; 0.917) algorithms performed the best. Regarding variable importance in the ALL model, the last HS, sex, BMI, and MUAC variables had the highest values. In the female model, these variables were HS, age, MUAC, and BMI, respectively. Conclusions: Machine learning algorithms have the ability to extract valuable insights from data structures, enabling accurate predictions for the early detection of sarcopenia. These predictions can assist clinicians in the context of predictive, preventive, and personalized medicine (PPPM).

## 1. Introduction

Sarcopenia is a common condition that affects muscle mass, increases morbidity due to fractures, decreases cognitive functions and increases the risk of cardiovascular disease, hospitalization, and mortality in older adults. It is a major contributor to frailty and disability and can significantly impact an individual’s quality of life. Early diagnosis of sarcopenia is important for the successful management and treatment of this condition [1,2,3,4].

In addition, sarcopenia causes an increase in direct and indirect health expenditures. Since the globally increasing elderly population will lead to more patients with sarcopenia, the identification and treatment of sarcopenia should be focused on reducing the financial burden of sarcopenia on health expenditures [5]. Thus, it is important to emphasize the earlier detection and therapeutic intervention for individuals prone to suffering with sarcopenia [6].

Traditionally, sarcopenia has been diagnosed using physical exams and functional assessments, such as measuring muscle strength and evaluating an individual’s ability to perform activities of daily living. However, these methods can be subjective and may not always accurately reflect an individual’s muscle mass and function.

Recently, machine learning models started to develop as a potential method of diagnosis of sarcopenia. Early diagnosis and treatment of sarcopenia can be achieved by using these developed and optimized models. Thus, while increasing the quality of life, it also reduces the burden on the health system.

The prevalence of sarcopenia depends on the criteria and cut-off values used to define sarcopenia; in addition, study groups consisting of the elderly living in a community, nursing homes, or hospitals may explain the large differences in the prevalence of sarcopenia. Regarding the new criteria of EWGSOP (European Working Group on Sarcopenia in Older People—EWGSOP 2), it has been reported that the prevalence of sarcopenia is 4.6–23.3% among the elderly living in a community [7,8,9,10].

The EWGSOP2 recommends the use of the SARC-F questionnaire as a way to elicit self-reports from patients on the signs that are characteristic of sarcopenia. The SARC-F can be readily used in community healthcare and other clinical settings [1]. In the meta-analysis examining the accuracy of the SARC-F as a screening test, although the screening sensitivity was low, its selectivity was high; therefore, it has been reported to be an effective tool to select participants that require further testing to diagnose sarcopenia [11].

It is recommended to use the SARC-F sarcopenia risk screening scale, which relies on self-reporting by elderly individuals and is a cost-effective and convenient tool to inquire about the characteristic signs of sarcopenia. The validity of the SARC-F in identifying individuals at risk of adverse outcomes associated with sarcopenia has been confirmed in various languages and communities, both within public health services and clinical practice [12,13,14,15,16,17,18].

Anthropometric measurements such as calf circumference (CC) and mid-upper arm circumference (MUAC) are inexpensive and easy-to-apply methods in the evaluation of muscle mass. A modified version of the SARC-F produced by adding calf circumference (CC) to the SARC-F (SARC-CalF) has significantly improved sensitivity and accuracy [16,19,20,21]. In a large cohort study, a low MUAC (<25 cm) was significantly associated with an increased risk of 15-year mortality in community-dwelling elderly persons [21].

In 2019, EWGSOP updated its definition of sarcopenia by introducing threshold values (EWGSOP2) and incorporated physical performance in assessing the severity of sarcopenia. Within the EWGSOP 2 diagnostic criteria, low muscle strength has become the primary factor in identifying sarcopenia, as it represents the most reliable measure of muscle function. When low muscle strength is detected in older individuals, it is categorized as ‘possible sarcopenia’, and the diagnosis of sarcopenia is confirmed through the presence of low muscle quantity/quality. If all the criteria of low muscle strength, low muscle quantity/quality, and low physical performance are met in older individuals, this indicates ‘severe’ sarcopenia [1].

Measuring handgrip strength (ESG) is a simple and cost-effective method. Low handgrip strength is associated with prolonged hospital stays, reduced quality of life, increased functional limitations, and mortality. Due to its ease of use, routine application in clinical practice and public health services is recommended [22,23,24]. In participants where handgrip strength cannot be measured, such as in individuals with conditions such as osteoarthritis or stroke, isometric torque methods are employed to assess lower extremity strength [25]. The 30 s chair stand test (sit-to-stand procedure) is another tool utilized to gauge muscle strength, specifically the strength of the quadriceps muscle group.

Among the tests used to assess physical performance, the Short Physical Performance Battery, walking speed, and timed up-and-go tests are the most commonly employed. “Walking speed” is recommended by EWGSOP 2 for evaluating physical performance due to its user-friendliness and ability to predict outcomes related to sarcopenia [1,26].

Calf circumference (WC) serves as an indicator of functional status in the elderly. Low WC is associated with disability and reduced functional capacity, while high WC is linked to a lower risk of frailty. Research has shown that a WC of <31 cm in both genders reflects health and nutritional status and predicts performance and survival in the elderly [27,28]. WC measurement can be employed as an indicator of muscle mass in sarcopenia screening, given its simplicity and widespread availability.

Mid-upper arm circumference (UMC) is an anthropometric measurement that reflects both muscle and fat mass when evaluating nutritional status. In a ten-year follow-up study conducted by Schaap et al. in the ‘Longitudinal Aging Study Amsterdam’, a decrease in UMC among elderly individuals with a baseline measurement of <31 cm was associated with increased mortality and reduced muscle mass due to sarcopenia [29]. Additionally, in the SarcoPhAge study, the mean UMC measurement in the sarcopenic group was significantly lower than that in the non-sarcopenic group [30].

There are numerous instruments used to evaluate the body’s muscle mass or fat tissue, each with varying levels of accuracy. Anthropometry offers the most cost-effective, portable, widely applicable, and non-invasive technique for assessing human body size, proportions, and composition. Given the limitations in clinical practice, where equipment for measuring body muscle mass (such as MRI, CT, DXA, and BIA) are constrained by factors such as high costs, lack of portability, exposure to low doses of radiation, and the need for trained personnel, anthropometric measurements provide a practical means to assess body composition and muscle mass. Utilizing these calculations, sarcopenia can be evaluated within public health services [26,31,32]. 

Considering the diagnostic criteria, there is a need for innovative support systems for the diagnosis of sarcopenia that are non-invasive, fast, cost-effective, suitable for community screenings, and can assist clinicians in situations where equipment such as the BIA and DXA are not possible to reach.

Ciu et al. (2020) [33] conducted a risk assessment for sarcopenia in patients with type 2 diabetes mellitus. They utilized the support vector machine (SVM) and random forest (RF) algorithms when dual-energy X-ray absorptiometry was not available. Their study involved 132 patients aged over 65 years. The sensitivity values ranging between 0.373 and 0.552 were obtained with the ML models [33]. Castillo-Olea et al. (2019) [34] assessed a dataset comprising 166 patients and 99 variables at the Tijuana General Hospital to develop an automated classifier for sarcopenia levels in older adults. They identified age, systolic arterial hypertension, mini nutritional assessment, the number of chronic diseases, and sodium levels as the five most crucial variables through feature selection. The RBF SVM classifier achieved the highest prediction performance, with an accuracy of 82.5%, an F1 score of 90.2, and a precision of 82.8% [34].

In this study, we aimed to develop an early diagnostic algorithm that includes some key measurements from the EWGSOP 2 diagnostic criteria commonly used for the potential diagnosis of sarcopenia in routine practice, along with patients’ demographic characteristics and physical measurements.

### Apply Machine Learning Methods for Sarcopenia 

Despite the rapid developments in medicine and technology in recent years, the specific mechanism of sarcopenia is still not fully explained. The use of successful predictive models to reveal hidden patterns in the data will support clinicians in early diagnosis and management of treatment processes. Machine learning can be defined as a method that recognizes and utilizes specific patterns in various types of data to make predictions. In some participants, it can uncover patterns that are not discernible to the human eye. Machine learning is rooted in the ability to learn from the data provided by researchers and mimics certain aspects of human cognition. Consequently, it falls within the realm of artificial intelligence (AI), which encompasses systems or machines capable of emulating human thought processes to accomplish specific objectives. When the studies in the literature are evaluated, there are a very limited number of ML methods used in the diagnosis of sarcopenia. In addition, newly developed algorithms with developing computer technologies are updated to better capture these patterns. In this study, the performances of the LightGBM, RF, and XGBoost models were evaluated for sarcopenia prediction by using 11 features determined in the datasets consisting of female + male (ALL) and female individuals in the workflow of PPPM (Figure 1).

## 2. Methods

### 2.1. Sample Size

The sample size of this study was calculated through an approach where the total population is unknown. Values at a 95% confidence level, 5% margin of error, and 11.8% of the population proportion were used for calculations, with a sample size of a minimum of 160 participants needed for this study. Considering the losses—by taking 20% more participants—210 participants were included in this study [35].

### 2.2. Study Design and Participants

This cross-sectional study was adopted with 210 participants who applied to the Ege University Faculty of Medicine, the geriatrics outpatient clinic for any reason, from July 2018 to May 2019. Community-dwelling participants aged 65 years and over were included in this study; written informed consent was obtained from all participants.

The exclusion criteria are as follows: an implanted pacemaker, severe heart failure, severe renal failure, severe mental illness, unable to walk, clinically visible edema, and unable to communicate.

### 2.3. Measurement of Muscle Mass, Muscle Strength, and Gait Speed

Body composition analysis was determined with electrical bioimpedance using the Tanita MC-780 multi-frequency segmental Body Composition Analyzer (Tanita Corporation, Tokyo, Japan). The appendicular skeletal muscle mass (ASM) was calculated with the Sergi equation for EWGSOP 2 criteria [36]. The appendicular muscle mass index (ASMI) was calculated based on the following equation: ASM (kg/height (m^2^)).

Muscle strength (kg) was assessed with the Takei Grip Strength Dynamometer^®^ (Takei, Saga, Japan). Handgrip strength (HS) measurements were made with the subjects in a sitting position, with the elbow and wrist in full extension, three times with an interval of five seconds on both hands, and the highest value among the measurement results was used for the analysis (30). Gender-specific cut-offs were used, with 27 and 16 kg in males and females for the EWGSOP 2 criteria [1], respectively. 

Usual gait speed (m/s) was performed by the subjects walking 4 m with usual speed and ≤0.8 m/s was defined as low walking speed.

### 2.4. Anthropometric Measurement

In all participants, height was measured using a stadiometer to the nearest 0.1 cm; weight was measured unclothed to the nearest 0.1 kg using a calibrated balance scale. Body mass index (BMI) was calculated using the weight (kg)/height (m^2^) equation.

The CC of participants was measured with an inflexible tape measure, in the sitting position, with the knee flexed to 90°, and the circumference of the widest part of the calf. Both the standard CC cut-off (<31 cm) and population-specific cut-off (<33 cm) were used [37]. MUAC was measured in a standing position, the midpoint of the participant’s left upper arm—located between the acromion and olecranon—was marked when the elbow bent to a 90° angle and was measured with an inflexible tape around the marked midpoint with the participant’s arm relaxed naturally. 

### 2.5. SARCF

The SARC-F is a 5-item self-reported questionnaire for screening sarcopenia risk (Appendix A). Responses are based on the patient’s perception of his or her limitations in strength, walking ability, standing up from a chair, stair climbing, and experiences with falls and is scored between 0 to 2 [15]. The SARC-F has proven validity in identifying individuals at risk for adverse outcomes associated with sarcopenia in a variety of languages and populations [11,12,14,15,17,38].

The EWGSOP2 (Figure 2) recommends the use of the SARC-F questionnaire as a way to elicit self-reports from patients on signs that are characteristic of sarcopenia. The SARC-F can be readily used in community healthcare and other clinical settings [1].

### 2.6. Statistical Analysis

Statistical analyses performed by using IBM SPSS V25 (IBM Corp. Released 2017. IBM SPSS Statistics for Windows, Version 25.0. Armonk, NY, USA: IBM Corp.) All machine learning analysis was performed with Automated Python Libraries on Azure Machine Learning Studio [39]. Categorical variables were calculated by the number of participants (n) and percentages (%). Continuous variables were presented by mean, standard deviation (std), minimum (min), maximum (max), and median (med) values. Normality assumption was tested using the Kolmogorov–Smirnov/Shapiro–Wilk test for continuous variables. Differences between continuous variables were compared using the Student’s *t* test/Mann–Whitney U test. Categorical variables were compared with the *X*^2^ test. Significance level was defined as *p* ≤ 0.05 for all statistical analyses.

Machine learning models (LightGBM, XGBoost, random forest, k-nearest neighbors, support vector machine) were used for differential diagnoses of sarcopenia and non-sarcopenia participants. We have evaluated predictive models by partitioning the original sample into a training set (80%) and test set (20%). ML models were trained 1000 times and results were then averaged to produce a single estimation. All groups (healthy/case, gender) were equally represented in training and test sets with their true proportion. Also, in this study, the emphasis was placed on the best results, with machine learning outcomes presented based on the most successful models.

### 2.7. Performance Measures

In this study, the performance of algorithms was evaluated with accuracy, AUC (area under the curve), F1 score, Matthews correlation coefficient (MCC), precision, and recall (Table 1) [40].

### 2.8. Variable Importance

Variable (feature) importance refers to the measure of the contribution or relevance of each variable or feature in a statistical or machine learning model for predicting or explaining the outcome of interest. It provides insights into the relative significance of different variables in influencing the model’s performance and helps in understanding the factors driving the observed patterns or trends in the data [41].

## 3. Results

### 3.1. Characteristics of Participants

A total of 210 community-dwelling participants aged 65 years and over were included in this study. Of those, 12 patients did not complete the SARC-F questionnaire and 8 patients lacked the measurement data of HS and were thus excluded from this study. Finally, the data from 190 participants were included in the analysis. 

In this study’s population, the median score of age in females was 71.0 years [65.0–88.0] and in males it was 74.0 years [65.0–90.0] (*p* = 0.005). The height average for females was lower than for males (*p* < 0.001). Conversely, BMI (*p* = 0.012) and the SARCF (*p* = 0.002) median was higher in females. The median MNA score was similar in males and females (*p* = 0.343). CC (*p* = 0.802) and walking speed (0.060) means had no statistical difference between males and females. The median HS score was found to be much higher in males (31.0 [18.0–43.0]) than in females (21.0 [6.0–41.0]) (*p* < 0.001). The mean MUAC was 29.5 ± 3.1 in males and 30.7 ± 3.7 in females, respectively (*p* = 0.029) (Table 2).

Smoking was found to be similar in men (10.3%) and women (6.8%) (*p* = 0.406). The distribution of the presence of hypertension showed a statistically significant difference according to sex (*p* = 0.019). The frequency in women (59.8%) was higher than in men (41.4%). Diabetes mellitus (*p* = 0.725) and malnutrition status (0.702) were also found to be similar according to sex. Prevalence of possible sarcopenia was 14.7% (12.9% in males, 19.0% in females, *p* = 0.276) using the EWGSOP-2 criteria (Table 3).

### 3.2. Variables and Diagnostic Performance of Machine Learning Algorithms

During the development of the ML models, variables were determined based on the EWGSOP2 criteria and the clinical expertise of the medical experts within the team.

In the ALL model, the LightGBM algorithm showed the highest performance with the values “Accuracy = 0.931; AUC = 0.975; Precision = 0.932 and Recall = 0.921”, while the SVM algorithm in the female model showed high prediction performance with the values “Accuracy = 0.939; AUC = 0.979” (Table 4, Figure 3). Hyperparameters and average compute time of the ML models were presented in Appendix A. Also, training set results were presented in Appendix A.

## 4. Discussion

Prevention and/or early treatment of muscle mass loss and/or sarcopenia not only improves the prognosis of patients but also reduces the financial burden on healthcare costs. Since the globally increasing elderly population will lead to the formation of more sarcopenic patients, the identification and treatment of sarcopenia or low muscle mass should be focused on in order to reduce the financial burden of sarcopenia on health expenditures [5].

Despite its negative effects on both quality of life and morbidity/mortality, and the burden it creates on health expenditures, the diagnosis of sarcopenia is difficult due to the high cost of the diagnostic methods, which are accepted as the gold standard, and are not available everywhere. 

The SARC-F, which was developed by Malmstrom and Morley in 2013 and whose validity studies have shown good results in various languages and populations, is a five-question sarcopenia screening scale based on the self-report of the elderly and is easy to answer [42]. However, the SARC-F only gives information about muscle function (strength and performance) without evaluating muscle mass. Although the relationship between sarcopenia and loss of muscle function is undeniable, evaluating only muscle function without identifying muscle mass causes a deviation from the EWGSOP’s definition of sarcopenia [43].

Anthropometric measurements such as calf circumference and upper-middle arm circumference are inexpensive and easy-to-apply methods for evaluating muscle mass, and when evaluated together with the SARC-F, which provides information about muscle function, it may provide us with the information about the presence of sarcopenia. Therefore, it can be used as a sarcopenia screening tool in health services and field studies where it is not possible to reach muscle mass assessment techniques such as the bioelectrical impedance analysis (BIA) or dual-energy X-ray absorptiometry (DXA), which are included in the EWGSOP criteria and which are accepted as the current gold standard for the diagnosis of sarcopenia.

In the participants where the equipment such as the BIA and DXA are not available for the diagnosis of sarcopenia, there is a need for methods that are non-invasive, fast, economical, can be used in population screening, and can support the clinician. The widespread use of biological data processing technology and the cheaper and widespread use of technologies have changed the direction of medical research. With the ML/DL algorithms which detect meaningful information from data structures, successful predictions have begun to be obtained that support clinicians in diagnosing many diseases. 

In our study, we aimed to develop an early diagnosis algorithm that includes some basic measurements in the EWGSOP 2 diagnostic criteria, which are routinely used for the diagnosis of possible sarcopenia, patients’ demographic characteristics, and physical measurements. We presented the performance of the models in predicting sarcopenia for ALL and females. The ALL model included 11 variables and the female model included 9 variables. We obtained an accuracy of 0.931 with the LightGBM algorithm via the ALL model and 0.939 with the SVM in the female model. The innovative algorithms used in this study showed different and higher performance than those used by the researchers in the literature.

Kang 2019 et al. [44] evaluated the features of 1698 male and 2322 female participants from the KNHANES results. A total of 17 risk variables were used for sarcopenia prediction with the following classification algorithms: logistic regression, support vector machine, gradient boosting, and random forest. They reported similar results (ROC-AUC) for either sex via machine learning algorithms (RF: 0.82 in men, 0.78 in women; support vector: 0.80 in men, 0.81 in women; gradient boosting: 0.81 in men, 0.81 in women; logistic regression: 0.82 in men, 0.80 in women). In our study, we obtained higher prediction performance by using 11 variables. Using the LightGBM algorithm, which is one of the new popular algorithms, we predicted an accuracy of 0.931 and an AUC of 0.975 [44].

Ciu 2020 et al. [33] made a risk assessment of sarcopenia in type 2 diabetes mellitus patients using the support vector machine (SVM) and random forest (RF) algorithms when dual-energy X-ray absorptiometry was not available. They evaluated 132 patients’ data who were over the age of 65 years and diagnosed them with T2DM in Changchun, China. Also, they selected some features out of over 40 features of patients’ data to train the SVM and RF classification models and regression models. They obtained higher performances with the SVM model than random forest (results of sensitivity between 0.373 and 0.546 with the RF and between 0.425 and 0.552 with the SVM). The number of participants in this study was similar to the number of participants in our study. On the other hand, researchers have obtained low model performances in their studies using classical machine learning approaches [33].

Castillo-Olea 2019 et al. [41] evaluated a dataset that included 166 patients and 99 variables in the Tijuana General Hospital to provide an automatic classifier of the level of sarcopenia in older adults using machine learning methods (k-nearest neighbors, linear support vector machines, radial basis function SVM, Gaussian process RBF, decision tree, random forest, multilayer perceptron, AdaBoost, Gaussian naïve Bayes, quadratic discriminant analysis) and to identify the importance of each variable. Using age, systolic arterial hypertension, mini nutritional assessment, number of chronic diseases, and sodium variables which were determined as the five most important variables as a result of the feature selection, the highest prediction performances (82.5% accuracy, 90.2% F1 score, and 82.8% precision values) were obtained using the RBF SVM classifier. They used boosting algorithms as well as classical machine learning approaches. We also obtained higher accuracy values using similar approaches in our study [41].

Ko B. 2021 et al. [45] developed a machine learning classification model (KNN, SVM, and NB) for predicting sarcopenia through an inertial measurement unit (IMU)-based physical performance measurement data of elderly females. To implement the sarcopenia prediction model, the data of 78 persons were used among the data of 105 persons in total and were based on the literature review. A total of 132 features were extracted from the collected data. They used five-fold cross-validation to assess the performance of a machine learning model. The k-nearest neighbors (KNN) algorithm classification model that uses 40 major features showed the best performance with a 0.881 value. However, all features used in the NB classification model showed the lowest performances with accuracy values of 0.830 for sarcopenia prediction. The female model in our study showed a higher performance than the latter study mentioned [45].

Agnes 2019 et al. [46] developed a binary logistic regression model for the prediction of sarcopenia. In their case–control study that included 104 older adults, the age, BMI, physical activity, grip strength, quadriceps strength, balance, and the SARC F showed a significant OR for sarcopenia. Despite having many similarities to our study, their analysis method was based on logistic regression, which forms the foundation of machine learning approaches. Unfortunately, the researchers have not presented the predictive performance of the model. The importance of including anthropometric measurements in the models in terms of the variables used has been emphasized. In this aspect, it is similar to our study [46].

Seok and Kim explored the impact of daily-life physical activity (PA) and obesity on sarcopenia prediction in their study. To demonstrate the feasibility of predicting sarcopenia using PA, they trained various machine learning models (including the gradient boosting machine, XGBoost, LightGBM, CatBoost, logistic regression, support vector classifier, k-nearest neighbors, random forest, multi-layer perceptron, and deep neural network) using the data samples from the Korea National Health and Nutrition Examination Survey. Among these models, the deep neural network (DNN) achieved the highest average accuracy of 81% when considering PA features across all data combinations. The accuracy increased to 90% with the inclusion of obesity-related information, such as total fat mass and fat percentage. Various metrics were used for precise model evaluation, and the algorithms exhibited significant predictive performance using only PA features, including waist circumference, with the AUC values being consistently around a value of 0.85 and often approaching or exceeding a value of 0.90. In our study, physical and anthropometric measurements were jointly evaluated, leading to high performances, which closely align with our models’ outcomes. Based on this studies’ results, it is evident that when appropriate anthropometric and physical measurements are collected from individuals, crucial information can be derived, significantly contributing to a definitive diagnosis within the models [47].

Due to advancements in computer technologies, the frequency of misdiagnoses in diseases is decreasing steadily. The utilization of high-performance algorithms for extracting precise information from complex patterns, such as deep learning, is on the rise in the healthcare sector. In a study conducted by Asaf et al., a deep learning model achieved a classification accuracy of 99.35% for breast cancer detection and classification. It is well-established that convolutional neural network architectures in deep learning algorithms exhibit exceptional predictive capabilities, particularly when imaging data are available. [48]. In another study, Ullah et al. achieved an impressive accuracy of 99.33% in classifying brain tumor MRI scans (to discriminate normal and brain tumors using a standard Kaggle brain tumor MRI dataset) using convolutional neural networks [49]. In our study, as we did not have access to imaging data and had a smaller sample size, we assessed our research using machine learning algorithms, yielding remarkably high performance. Here, the importance of variable selection in machine learning models also becomes evident. 

As observed in the literature, the development of models for sarcopenia diagnosis continues to progress. The individuals included in our study were patients who voluntarily sought care at a university hospital. Therefore, the prevalence rate aligns with the general population. We explored models that encompassed all participants and separately examined female individuals. Unfortunately, due to the limited number of male participants, separate models for men could not be evaluated. While the number of participants in our study is similar to some studies in the literature, the prevalence of the disease underscores the importance of conducting dedicated screening efforts, particularly in nursing homes and areas with denser elderly populations.

The models we developed in our study demonstrated high performance, providing valuable support to clinicians in diagnosing sarcopenia. Innovative machine learning algorithms such as XGBoost and LightGBM have proven to be highly effective in identifying patterns within the disease data for the diagnosis of challenging conditions. According to our results, the development and implementation of models suitable for rapid, cost-effective population screenings, without the need for devices such as the BIA and DXA, which are costly and expose patients to radiation (even at low levels), are crucial for alleviating the burden on healthcare systems.

### 4.1. Key Points

These results highlight that ML algorithms can support clinicians in the early diagnosis of sarcopenia without using the BIA and/or DXA.Early diagnosis and therapeutic intervention for individuals prone to suffering sarcopenia decrease the risks of disability, hospitalization, and mortality.Early diagnosis of sarcopenia models will have a significant economical contribution to the healthcare system.

### 4.2. Study Limitations

The first limitation is that the muscle mass for sarcopenia diagnosis was measured using the BIA—which can be influenced by the hydration status. Secondly, only ALL participants and female results were presented in this study. The sample was taken from spontaneous admissions from a university hospital, and the prevalence of sarcopenia in this study reflects community-dwelling elderly persons. Therefore, the ML models could not be applied due to the low number of male participants. 

Even if we analyzed “ALL and only female” participants, as we can see from Table 4, ‘sex’ is an important feature in prediction models. Therefore, we recommend to train ML models for male and female participants individually.

There is a need for studies in which participants are recruited from centers such as nursing homes where the frequency of sarcopenia is higher.

## Figures and Tables

**Figure 1 healthcare-11-02699-f001:**
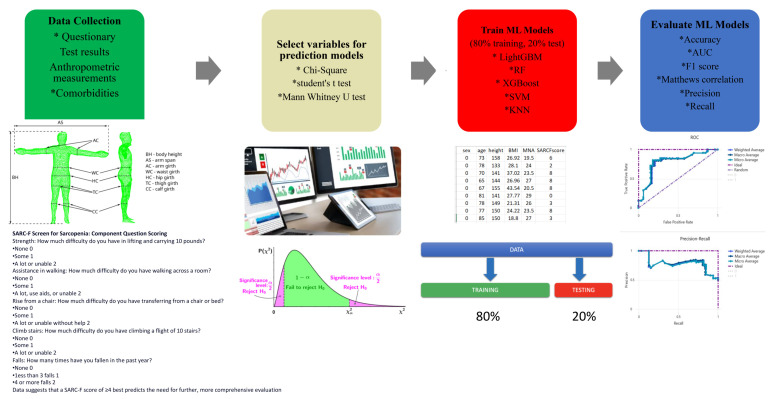
Study workflow.

**Figure 2 healthcare-11-02699-f002:**
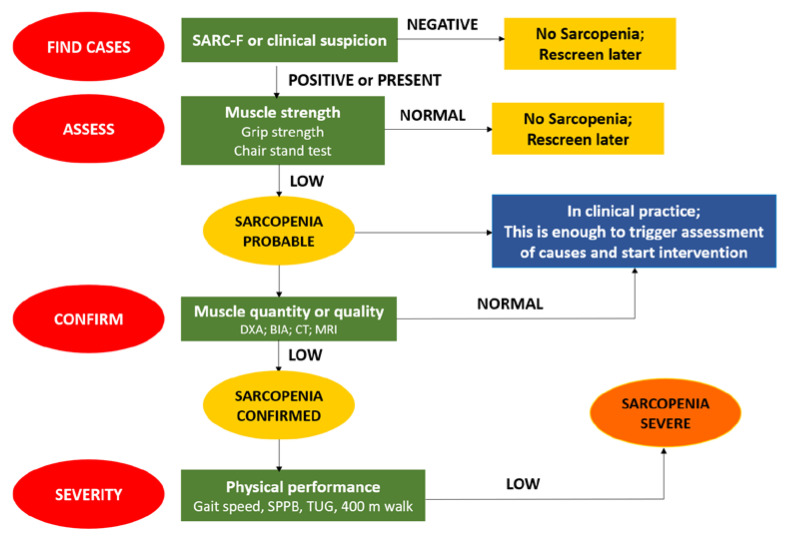
The EWGSOP2 algorithm for diagnosis of sarcopenia.

**Figure 3 healthcare-11-02699-f003:**
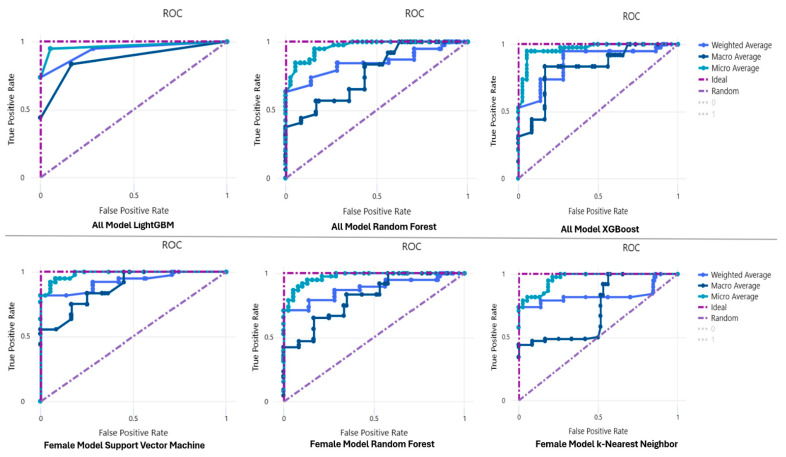
Assessment of sarcopenia prediction model performance using AUC-ROC curves. RF: random forest; SVM: support vector machine; KNN: k-nearest neighbors. Variables importance values of the models are presented in Table 4. In the ALL prediction model, HS (1.071), sex (0.839), and BMI (0.328) variables have the highest importance in the model. Also, HS (1.798), age (0.631), and MUAC (0.336) were the most important variables for the female prediction model (Table 5).

**Table 1 healthcare-11-02699-t001:** Performance measurements of the models.

$Accuracy=\frac{\left(TP\;+\;TN\right)}{\left(TP\;+\;TN\;+\;FP\;+\;FN\right)}$ $Precision=\frac{TP}{\left(TP+FP\right)}\;\;\;\;\;\;\;\;Recall=\frac{TP}{\left(TP+FN\right)}$ $F1\;score=2\times \frac{Precision\;\times \;Recall}{Precision\;+\;Recall}$ $MCC=\frac{TN\;\times \;TP\;-\;FN\;\times \;FP}{\sqrt{\left(TP\;+\;FP\right)\;\;\left(TP\;+\;FN\right)\;\;\left(TN\;+\;FP\right)\;(TN\;+\;FN)}}$	**Diagnostic Test**	**Gold Standard**	
		**Positive**	**Negative**
	Positive	TP	FP
	Negative	FN	TN
	TP: true positive; TN: true negative; FN: false negative; FP: false positive.		

**Table 2 healthcare-11-02699-t002:** Comparisons of anthropometric measurements according to sex.

Variables	Sex	Mean ± SD	Med [Min–Max]	Test Statistics; *p*-Value
Age (year)	Female	72.4 ± 5.6	71.0 [65.0–88.0]	U = 2848.5 *p* = 0.005
	Male	75.4 ± 6.7	74.0 [65.0–90.0]	
Height (cm)	Female	153.1 ± 6.3	153.0 [133.0–172.0]	*t* = −10.789 *p* < 0.001
	Male	165.6 ± 7.8	166.5 [142.0–179.0]	
BMI (kg/m^2^)	Female	29.4 ± 5.5	29.1 [16.6–47.7]	U = 2951.0 *p* = 0.012
	Male	27.3 ± 4.3	26.6 [18.6–39]	
MNA score	Female	25.3 ± 3	26 [13–30]	U = 3471 *p* = 0.343
	Male	25.5 ± 3.3	26.5 [12.5–29.5]	
SARC-F score	Female	2.9 ± 2.3	2.0 [0.0–9.0]	U = 2742.0 *p* = 0.002
	Male	1.7 ± 1.7	1.0 [0.0–6.0]	
Handgrip strength (HS) (kg)	Female	21.1 ± 5.5	21.0 [6.0–41.0]	U = 957.0 *p* < 0.001
	Male	30.9 ± 6.3	31.0 [18.0–43.0]	
Walking speed m/s	Female	1.0 ± 0.3	1.0 [0.1–1.8]	*t* = −1.893 *p* = 0.060
	Male	1.1 ± 0.3	1.1 [0.4–1.8]	
Calf circumference (cm)	Female	37.9 ± 4.2	38.0 [26–48]	*t* = −0.252 *p* = 0.802
	Male	38 ± 3.6	38 [32–45]	
Mid-upper arm circumference (MUAC) (cm)	Female	30.7 ± 3.7	30.5 [21–43]	*t* = 2.197 *p* = 0.029
	Male	29.5 ± 3.1	29 [24–38]	

U: Mann–Whitney U test, *t*: Student’s *t* test, *p* < 0.05: significance level.

**Table 3 healthcare-11-02699-t003:** Comparisons of categorical variables according to sex.

Variables		Sex		Total n (%)	Test Statistics; *p*-Value
		Female n (%)	Male n (%)		
Smoking	Yes	9 (6.8)	6 (10.3)	15 (7.9)	*X*^2^ = 0.689 *p* = 0.406
	No	123 (93.2)	52 (89.7)	175 (92.1)	
Hypertension	Yes	79 (59.8)	24 (41.4)	103 (54.2)	*X*^2^ = 5.537 *p* = 0.019
	No	53 (40.2)	34 (58.6)	87 (45.8)	
Diabetes Mellitus	Yes	31 (23.5)	15 (25.9)	46 (24.2)	*X*^2^ = 0.124 *p* = 0.725
	No	101 (76.5)	43 (74.1)	144 (75.8)	
Malnutrition status	Yes	5 (3.8)	3 (5.2)	8 (4.2)	*X*^2^ = 0.192 *p* = 0.702
	No	127 (96.2)	55 (94.8)	182 (95.8)	
Sarcopenia	Healthy	115 (87.1)	47 (81.0)	162 (85.3)	*X*^2^ = 1.188 *p* = 0.276
	Possible sarcopenia	17 (12.9)	11 (19.0)	28 (14.7)	

*X*^2^: Chi square test; *p* < 0.05: significance level.

**Table 4 healthcare-11-02699-t004:** Sarcopenia prediction models (test set).

ALL Model	LightGBM	RF	XGBoost
Accuracy	0.931	0.894	0.867
AUC	0.975	0.937	0.927
F1 score	0.921	0.871	0.852
Matthews correlation coefficient	0.676	0.544	0.533
Precision	0.932	0.885	0.863
Recall	0.921	0.892	0.874
**Female Model**	**SVM**	**RF**	**KNN**
Accuracy	0.939	0.924	0.917
AUC	0.979	0.958	0.953
F1 score	0.918	0.934	0.927
Matthews correlation coefficient	0.679	0.782	0.693
Precision	0.934	0.952	0.914
Recall	0.929	0.953	0.857

RF: random forest; SVM: support vector machine; KNN: k-nearest neighbors.

**Table 5 healthcare-11-02699-t005:** Variable importance of ALL and female models.

Variable Importance of Models			
ALL Model		Female Model	
Variable	Importance	Variable	Importance
Handgrip strength **	1.071	Handgrip strength **	1.798
Sex	0.839	Age	0.613
BMI	0.328	Mid-upper arm circumference	0.336
Mid-upper arm circumference	0.279	BMI	0.171
Walking speed **	0.243	SARCF **	0.123
HT	0.172	HT	0.119
DM	0.155	Walking speed **	0.109
Age	0.134	MNA	0.068
Calf circumference	0.118	Calf circumference	0.041
SARCF **	0.085		
MNA	0.030		
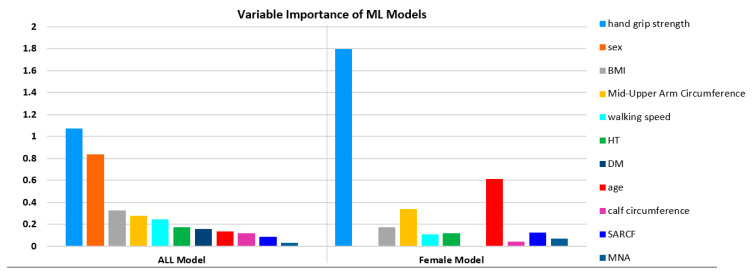			

The variables in the ‘EWGSOP2 algorithm for the diagnosis of sarcopenia’ are presented in the table with the ** symbol.

## Data Availability

Sharing this data with researchers is only possible with the approval of the ethics committee since this dataset contains real clinical data. Therefore, if researchers interested in the data contact the corresponding author, the dataset will be shared in accordance with data privacy rules.

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
