# Peer review of "Performance Evaluation of Machine Learning Algorithms for Sarcopenia Diagnosis in Older Adults"

_healthcare, 2023, doi:10.3390/healthcare11192699_

Round 1

Reviewer 1 Report

The detailed comments on the proposed approach are enlisted in the attached document. Although the authors have written a sound and thorough approach, some comments are presented below to improve the quality of the manuscript. 

Author Response

Dear Reviewer 1,

Thank you very much for your feedback and perfect contribution to our work.  This process has provided us with the opportunity to rectify several aspects that may have been overlooked during the initial drafting and submission of the article. We hope that we have adequately addressed all your questions.

In the revised document, I have highlighted all the sections where changes were made using yellow and green colors. I crossed out the parts that needed to be deleted, they are still in the text

Once again, we extend our gratitude for your time and feedback.

Author Response

Dear Reviewer,

Thank you very much for taking the time to provide feedback on our work. We hope that we have adequately addressed your questions and concerns to your satisfaction. In our revisions, we have not removed any content but have instead crossed out the expressions we intended to eliminate. Newly added sections are highlighted in yellow. We trust that this version of the article provides greater clarity and insight.

Best regards.

Round 2

Reviewer 1 Report

Thank you for incorporting many of the reviews. However, i notice that there are few papers in the references where they have also used ML for the similar approaches, you should compare your approach with them in term of accuracy, algorithm used and other factors stating why your approach is better than theirs.

Moreover, when discussing the future sections, deep learning and its application may be discussed to further improve the accuracy of the proposed approach. 

·       The references may be used when writing the rewriting the discussion section of the paper.

DeepBreastCancerNet: A Novel Deep Learning Model for Breast Cancer Detection Using Ultrasound Images, 

Detection of sarcopenia using deep learning-based artificial intelligence body part measure system (AIBMS)

A Robust End-to-End Deep Learning-Based Approach for Effective and Reliable BTD Using MR Images

Author Response

Thanks for your recommendation. We discussed and added the papers in below to the manuscript. We did our best to define differences and similarities between our study and these papers. We highlighted the gain on accuracy of our study but they have not reported any optimization on algorithms’ parameters. Therefore, it is almost impossible to compare our results with their results. The another blocker on comparison is they have not reported the compute times which is important for cost/benefit situation of the studies.

PS: We have marked all the changes made this time in blue.

A Robust End-to-End Deep Learning-Based Approach for Effective and Reliable BTD Using MR Images

DeepBreastCancerNet: A Novel Deep Learning Model for Breast Cancer Detection Using Ultrasound Images, 

Author Response

Thanks for your recommendation about the selecting subset or creating augmented dataset. This idea is definitely reasonable for model building or training. Our study’s major contribution and value is using real world data with real caveats (unequal group distribution, small sample size etc.). If we add synthetic scenarios to the manuscript, we may lose the attention of the clinicians and overall study structure. Therefore, we added training set results as a supplementary table but not added subset/augmented data set results to the manuscript.

PS: We have marked all the changes made this time in blue.
